# Determination of Nutrients in Liquid Manures and Biogas Digestates by Portable Energy-Dispersive X-ray Fluorescence Spectrometry

**DOI:** 10.3390/s21113892

**Published:** 2021-06-04

**Authors:** Michael Horf, Robin Gebbers, Sebastian Vogel, Markus Ostermann, Max-Frederik Piepel, Hans-Werner Olfs

**Affiliations:** 1Abteilung Technik im Pflanzenbau, Leibniz-Institut für Agrartechnik und Bioökonomie (ATB), Max-Eyth-Allee 100, D-14469 Potsdam, Germany; rgebbers@atb-potsdam.de (R.G.); svogel@atb-potsdam.de (S.V.); 2Fachbereich 1.4, Bundesanstalt für Materialforschung (BAM), Zweiggelände Adlershof, Richard-Willstätter-Straße 11, D-12489 Berlin, Germany; markus.ostermann@bam.de; 3Fachgebiet Pflanzenernährung und Pflanzenbau, Fakultät Agrarwissenschaften und Landschaftsarchitektur, Hochschule Osnabrück, Am Krümpel 31, D-49090 Osnabrück, Germany; m.piepel@hs-osnabrueck.de (M.-F.P.); h-w.olfs@hs-osnabrueck.de (H.-W.O.)

**Keywords:** handheld XRF, animal slurry, organic fertilizers, fertilization management, precision farming

## Abstract

Knowing the exact nutrient composition of organic fertilizers is a prerequisite for their appropriate application to improve yield and to avoid environmental pollution by over-fertilization. Traditional standard chemical analysis is cost and time-consuming and thus it is unsuitable for a rapid analysis before manure application. As a possible alternative, a handheld X-ray fluorescence (XRF) spectrometer was tested to enable a fast, simultaneous, and on-site analysis of several elements. A set of 62 liquid pig and cattle manures as well as biogas digestates were collected, intensively homogenized and analysed for the macro plant nutrients phosphorus, potassium, magnesium, calcium, and sulphur as well as the micro nutrients manganese, iron, copper, and zinc using the standard lab procedure. The effect of four different sample preparation steps (original, dried, filtered, and dried filter residues) on XRF measurement accuracy was examined. Therefore, XRF results were correlated with values of the reference analysis. The best R^2^s for each element ranged from 0.64 to 0.92. Comparing the four preparation steps, XRF results for dried samples showed good correlations (0.64 and 0.86) for all elements. XRF measurements using dried filter residues showed also good correlations with R^2^s between 0.65 and 0.91 except for P, Mg, and Ca. In contrast, correlation analysis for liquid samples (original and filtered) resulted in lower R^2^s from 0.02 to 0.68, except for K (0.83 and 0.87, respectively). Based on these results, it can be concluded that handheld XRF is a promising measuring system for element analysis in manures and digestates.

## 1. Introduction

During the last three decades, high nitrate concentrations in groundwater have been identified as a serious problem for clean drinking water. In many regions across Europe, this is mainly caused by an oversupply of nitrogen (N) bound in animal manures or biogas digestates on agricultural fields [1,2]. Besides nitrogen, phosphorus (P) can also cause problems leading to eutrophication of non-agricultural ecosystems if it is transported via surface runoff or soil erosion into surface water bodies [3,4]. Furthermore, long term accumulations of zinc (Zn) and copper (Cu) (mainly occurring in pig slurries) can contaminate soils and lead to environmental problems due to toxic effects on soil’s microbiology and plants at higher concentrations [5,6]. To minimize negative impacts on non-agricultural ecosystems, the European Union (EU) forced Germany to reduce nitrogen and phosphorous emissions from agriculture. However, the EU guidelines to limit application rates of manure on agricultural fields can only be implemented correctly if all relevant nutrient concentrations are quantitatively known and not estimated based on empiric values from recommendation tables. Such tabulated values cannot give reliable data for all different types of liquid organic manures because nutrient compositions depend on many different factors like animal species and age, feeding stuff compositions, storage managements, etc. [7].

Due to their heterogeneous character, slurry usually separates into solid and liquid phases with varying chemical compositions in time and space. Consequently, storage tanks filled with liquid organic manures have to be homogenised carefully to ensure a representative sampling. This homogenisation is a laborious step, which is in general done right before field application. After intensive homogenisation, a sub-sample is shipped to a lab for nutrient analysis. However, this traditional chemical analysis is cost and time consuming. While reliable on-farm methods have been developed for ammonium concentrations in liquid manures, e.g., [8], no such quick-tests are available for other nutrients contained in organic fertilizers. Hence, rapid and reliable in-situ methods are needed especially for analysing large numbers of samples to achieve more representative characterizations of slurries.

X-ray fluorescence (XRF) spectrometry is a well-established technique to determine the elemental composition of specimens, e.g., in the metal working, glass and cement industry [9,10]. It is also used for the investigation of archaeological objects and artwork [11]. Several studies show that XRF spectrometry can be also utilized for characterizing samples of different environmental media such as rocks [12], soils [13], sediments [14] or plant materials [15].

XRF spectrometers exist either as lab-based (floor-standing or benchtop) or as handheld devices. Lab XRF spectrometers work with a rather complex technical setup and are mostly equipped with a wavelength dispersive system (WDXRF). In contrast, handheld units are generally based on an energy dispersive system (EDXRF). WDXRF instruments are characterized by a higher resolution and a lower detection limit but lower measurement speed, often measuring only one element per analytical run. Handhelds with EDXRF are generally smaller, less expensive and faster than WDXRF systems, measuring a wide range of elements at once. However, EDXRF instruments cannot reach the precision of WDXRF spectrometers, especially for lighter elements in a matrix consisting of mainly heavy elements [16], whereas EDXRF benchtop instruments have advantages in measuring heavy elements in a light matrix (with a low z).

To produce XRF radiation, a sample is irradiated with high energy X-rays (called primary X-rays) in the wavelength range of 0.01–10 nm (equivalent to 125–0.125 keV). As a consequence, some atoms get ionized by ejecting an electron from one of the inner orbitals and an electron from one of the outer, higher energy orbitals fills the produced vacancy (Figure 1). Due to the transition to a lower energy state, the electron releases photons with an energy equal to the specific difference in energy of the two involved orbitals. This kind of radiation is called “secondary X-rays” or “X-ray fluorescence” [9]. It is unique to each element leading to characteristic emission spectra. By detecting the amount and frequency of the secondary radiation, the elementary composition of a sample can be qualitatively and quantitatively determined. The most pronounced emissions are generated by transitions from L to K shells (K_α_) and from M to K shells (K_β_) of the atom [16]. On the one hand, the intensity of these fluorescence emissions depends on the energy of the incoming X-rays. In general, the lower the incident X-ray energy, the higher is the absorption cross section and the higher is the fluorescence intensity. However, if the X-ray energy is intensified approximating the binding energy of an inner electron, a strong increase in absorption cross section will be recognized slightly above the electron’s binding energy (K- or L-edge). On the other hand, the fluorescence intensity is reduced when secondary X-rays are either absorbed by an electron on their way out of the atom emitting another electron instead (called Auger-electron) [9], by air molecules, or window-materials on their way to the detector cell. In summary, the ratio of produced vacancies and emitted fluorescence photons is called fluorescence yield and increases with the atomic number of an element. This means that lighter elements (e.g., carbon or nitrogen) are more difficult or even impossible to be analysed depending on the XRF system. As WDXRF systems are more sensible than EDXRF systems, elements from beryllium (z = 4) to uranium (z = 92) can be detected [9], whereas EDXRF systems can only detect elements from sodium (z = 11) to uranium. Field portable XRF systems cannot analyse elements lighter than magnesium (z = 12) [17].

However, until now, only few publications exist examining the use of XRF for manure analysis and publications for biogas digestates are completely missing. Dao and Zhang [18] investigated poultry litter, Roa-Espinosa et al. [19] examined dairy manure, [20] Weindorf et al. focused on composted dairy manure, two Japanese working groups published results for a mixed manure sample set (dairy, pig and poultry) [21,22], and Sapkota et al. [23] dealt with dairy and poultry manure. They all achieved very promising results with R^2^ between 0.80 and 0.99 for most of the plant nutrients.

The present study aims at evaluating the performance of a portable EDXRF spectrometer in measuring important plant nutrients, i.e., phosphorus (P), potassium (K), magnesium (Mg), calcium (Ca), sulphur (S), manganese (Mn), iron (Fe), copper (Cu), and zinc (Zn) in liquid manures and biogas digestives. Furthermore, the effect of different sample preparation methods on XRF results was tested and the instrument’s limit of quantification (LOQ) for each selected element in liquid and dry manure samples was determined. An additional objective of this study was to examine if XRF results of the three sample types (i.e., pig manure, cattle manure, biogas digestate) differ in accuracy.

## 2. Materials and Methods

### 2.1. Sample Set and Sample Preparation

For the present study, a set of 62 liquid manure and biogas digestate samples was used. The samples were collected in Northwest Germany comprising 41 pig manure (18 hog, 19 sow, 4 piglet), 11 cattle manure (8 dairy, 3 cattle), and 10 digestate samples. The sample material was intensively homogenised using a stainless steel mixer (Blender CB15VXE, Waring Commercial, Torrington, CT, USA), and were stored in 0.5 L plastic bottles at −18 °C.

### 2.2. Reference Analysis

As reference analysis, the official standard laboratory method based on acid digestion in a microwave system followed by inductively coupled plasma—optical emission spectrometry (ICP-OES) was applied with a well-proven modified version of DIN EN ISO 11885:2009 [24] at LUFA Nord-West (Hameln, Germany), a certified laboratory for slurry analysis. The elemental concentrations are expressed on a fresh weight basis. Dry matter content (DM) was determined with DIN EN 12880:2001 [25].

### 2.3. Sample Preparation for XFR Measurement

For the XRF measurements, samples were analysed with four different preparation steps after thawing over night at room temperature:(1)no preparation (original),(2)drying at 65 °C in a drying oven for at least 12 h and grinding with a mortar (dried),(3)filtration to <100 µm (filtrate), and(4)drying and grinding of the filter residues >100 µm (dried filter residues).

To analyse these four sample sets with the XRF spectrometer, the liquids or dried powders were filled in a 32 mm XRF plastic sample cell with a polypropylene X-ray film at the bottom (XRF sample cup [SC-4331] with polypropylene X-ray film circles [TF 240-255], FluXana GmbH & Co KG, Bedburg-Hau, Germany). All samples were measured twice in different positions by rotating the sample cell by about 90° and the average was calculated for each element.

### 2.4. XRF Fluorescence Analysis

XRF analysis was conducted with a field portable X-ray-fluorescence spectrometer (Niton XL3t Ultra 955 Hybrid, Thermo Fisher Scientific, Waltham, MA, USA) equipped with an EDXRF system. The silicon drift detector with geometrically optimized large drift detector technology (GOLDD+) attains a resolution of about 160 eV at 53,000 counts per four seconds shaping time. The spectrometer is equipped with an X-ray tube consisting of a silver anode (6–50 kV, 0–200 µA). For this study, it was focused on the elements P, K, Mg, Ca, S, Mn, Fe, Cu, and Zn as target parameters because of their relevance as plant nutrients. Unfortunately, nitrogen as an essential plant nutrient is not in the measurement range of the used portable XRF spectrometer.

For measurements, the internal standard modus ‘TestAll Geo’ was selected, which uses both the ‘Fundamental Parameter’ and the ‘Compton Normalization’ calibration, respectively. Furthermore, the instrument offers four different predefined voltage, current and filter settings (light, low, main and high), specialized for measuring different groups of elements by excluding photons of different energy ranges. Thus, the detecting range is focusing on the photon energy range of selected elements for a chosen time, improving the signal quality. In this case, three of these predefined settings (light, low and main) were used and measurement times were set to 30 s for each filter setting, resulting in a total measurement time of 90 s for a single analytical run for all target elements. The high filter could be deactivated because the target elements were not in the corresponding energy range.

To improve the accuracy of an XRF spectrometer, it is a common approach to calibrate the instrument manually with standards, consisting of the same or a similar matrix as the samples of interest. However, in this case, a manual calibration for liquid organic manure samples could not be achieved because the instrument required higher element concentrations for the calibration range than those of the available samples, of which 10 were supposed to be used as calibration reference standards. Nevertheless, the correct measurement setup of the spectrometer was checked with two certified reference materials NIST SRM 2780 (agricultural soil) and 2709a (hard rock mine waste).

To determine device and sample specific limits of detection for the utilized handheld XRF analyser, the lowest measured concentration for each element and for each sample preparation step was defined as the instrument’s LOQ, but only in the case of further occurring concentrations that were too low to be detected.

### 2.5. Statistical Analysis

Concentrations based on the XRF measurements of the four sample preparation sets were correlated with reference ICP-OES concentrations using a univariate linear regression model for each element. As quality criterion for each correlation, the coefficient of determination, the bias and the slope were chosen. All calculations were conducted with RStudio [26], a free software environment for statistical computing and graphics with R [27].

## 3. Results

Table 1 shows the descriptive statistics of dry matter (DM) content and nine element concentrations for all 62 liquid manure and biogas digestate samples determined by standard lab procedures. The examined samples have typical concentrations found in organic fertilizers [28,29] and show high variation coefficients from 31 to 48% for most of the constituents and even higher values for Fe (84%) and Cu (108%). This wide range of element concentrations made the data set particularly suitable for examining the instrument’s capability to analyse liquid manures and digestates precisely and accurately.

Table 2 summarizes the coefficients of determination for element concentrations obtained by ICP-OES and XRF depending on the four different sample preparation procedures. Best regressions for each element are additionally visualized later on in Figure 3. The regressions of Mg and Mn for liquid samples (original and filtrate) could not be calculated because most of the sample concentrations were below the instrument’s LOQ. In comparison to the liquid samples (original and filtrate), results for all elements (except K) of dried samples show a clear tendency of improving regression models due to drying. For example, the coefficient of determination for S in liquid samples (original and filtrate) was poor with 0.02 and 0.04. Dried samples (dried and dried filter residues) showed much better R^2^s of 0.69 and 0.66, respectively. In general, dried samples gave best results for most of the elements (Mg, P, S, Ca, Mn). For the heavier elements (Fe, Cu, Zn), dried filter residues samples had slightly better R^2^s than dried samples. As mentioned above, only potassium showed a better correlation in liquid samples with R^2^s of 0.83 and 0.87 for fresh samples and filtrates. One reason might be that potassium mainly exists as solved and hydrated K^+^ ions in the liquid phase, while all other target elements predominantly occur in the solid phase of the liquid fertilizer [30].

Three effects can mainly explain the general improvement of XRF results for dried samples. First, the presence of water leads to scatter effects of primary X-rays and a higher absorption of secondary X-rays [31]. Second, dried samples have higher elemental concentrations caused by an average 18-fold increase during the drying process. Third, element concentrations of organic fertilizers are often correlated to dry matter (DM) contents (Table 3), which means that the increase factor by drying is higher for those samples having lower element concentrations. For example, K correlates well with DM having a Pearson coefficient (r) of 0.77. The lowest DM concentration is 0.59% (Table 1) and the corresponding K concentration of this sample is 0.04%, which is also the lowest K concentration of the sample set. The calculated enriched K concentration after drying is 6.8% (due to an increase drying factor of 169), which is even slightly higher than the calculated enriched K concentration (6.7%; increase drying factor of 13) of the sample with the originally highest K concentration (0.52%). Thus, especially those samples with lower concentrations are enriched with a higher factor implying a lower probability to fall below the limit of quantification. For this reason, drying leads to a significant improvement of quantifying low nutrient concentrations. In the case of K, average concentrations in dried samples are slightly higher than in wet samples (data not shown). Nevertheless, R^2^s for K are still higher in liquid than in dried samples. The only plausible explanation for this is a more homogenous distribution in liquid samples in comparison to dried samples leading to a higher accuracy in analysis.

Higher correlations, found between P, Mg, Ca, Mn, and Zn, ranged from 0.73 to 0.85 (Table 3). Potassium and sulphur correlated best with dry matter content (r = 0.77 and 0.69), whereas Fe and Cu neither showed any strong correlations to other elements nor to DM.

In Figure 2, three raw XRF fluorescence spectra of a pig manure sample are exemplarily visualized, recorded by the spectrometer for each used filter setting (light, low, main). In each of the spectra, differently intensified peaks of various regions, caused by the three filter settings, can be identified for an enhanced quantitative evaluation. In the left spectrum, light filter settings especially intensify peaks of light elements like Mg, P and S in the region of about 1–3 keV, whereas the peak for Mg is still very small signifying the difficulty to accurately quantify this light element. In the middle spectrum with low filter settings, peaks of relevant elements like Mn and Fe from 5–8 keV are intensified, whereas K and Ca show a high counting rate similar to the left spectrum. In the right spectrum with main filter settings, especially relevant peaks of Cu and Zn are intensified in the region of 8–10 keV, whereas peaks of the light elements like Mg cannot be identified in the noise of that spectrum. The internal analysing programmes of the spectrometer use these spectra to calculate element concentrations. Before measuring samples of liquid organic manures, the measurement accuracy of the spectrometer was successfully proved with two certified reference materials, while using the analysing mode ‘TestAll Geo’.

Best correlations between the data determined by the reference standard lab procedure and XRF concentrations are visualized in Figure 3. The bias is almost zero in all cases. For P, S and Mn, the slope is close to 1.0. Magnesium and S show smaller slopes of 0.91, whereas Ca, Cu, and Zn have higher slopes of about 1.4. Slopes are highest for Fe (2.0) and K (2.3), respectively (Figure 3). It is suggested to calibrate the XRF spectrometer with these biases and slopes for each element when analysing samples with similar matrices.

Magnesium is the lightest element that can be detected and quantified by the handheld XRF spectrometer used in this study. However, Mg produces weaker signals than other elements. Thus, for the sample set in this study only higher concentrations above 0.12% in liquid samples and above 0.4% in solid samples could be quantified and much higher standard deviations in comparison to other elements were found. The different LOQs of about 0.12% and 0.4% show that the instrument’s LOQ depends on the sample matrix. Matrices of liquid samples (original and filtered) as well as solid samples (dried and dried residues) were very similar with almost equal LOQs. Thus, only the average LOQs for the two subsample sets (liquid and solid) were published.

Besides Mg, Mn in liquid samples could neither be quantified in many samples due to lower Mn concentrations and a higher LOQ of about 30 ppm in liquid samples in comparison to LOQs of Fe (about 20 ppm), Cu (about 8 ppm) and Zn (about 7 ppm). Furthermore, the standard deviations for Mn were also higher compared to the other elements. For S, the LOQ for liquid samples was about 0.02%. LOQs for P, K and Ca in liquid samples were not identified, because all sample concentrations exceeded the unknown LOQ. Target element concentrations of all dried samples did not exceed the LOQ and thus could be quantified well except the amount of Mg in 28 dried samples. This resulted in a regression of only 34 dried samples for Mg concentrations (Figure 3).

A further aspect concerning the reference method should be taken into consideration: ICP-OES is known for its good precision and accuracy with low chemical interferences due to the high plasma temperature above 6000 °K and the inert argon gas [32]. It is a suitable conventional method to be selected for reference analysis because of its reliable and standardized procedures. However, matrix-depending and spectral interferences may occur and have to be considered carefully. Furthermore, measured mass concentrations do not always represent the total concentrations for a specific element depending on the chosen chemical digestion procedure for the samples (e.g., microwave-assisted digestion with sulfuric acid versus aqua regia digestion). In contrast, XRF concentrations are always total concentrations, which means that XRF values might be a little bit higher compared to ICP-OES concentrations. Nevertheless, this presumably small effect cannot explain high slopes over 1.4 as found for Ca, Cu and Zn. Most likely, the reason for such high slopes are matrix-depending effects that are specific for each element. Examining the three different types of organic fertilizers (pig manure, cattle manure, biogas digestate) there is no obvious deviation in quantifying elemental concentrations via XRF spectrometry. Nevertheless, concentrations in dried samples tend to be slightly increased for cattle samples and slightly decreased for digestate samples, which is most evident for Ca (Figure 3). The reason for this effect is probably due to small differences in the solid sample matrix, which could not be identified when analysing liquid samples (e.g., K in filtrated samples), probably because of a more homogenous structure. Furthermore, K concentrations show higher values in cattle and digestate than in pig samples, which coincides with empiric values [28,29].

Table 4 shows a comparison of R^2^ values from XRF and reference concentrations of selected elements found in the literature. Roa-Espinosa et al. [19] observed almost perfect R^2^s close to 1.0 for all these elements in dairy manure. However, in contrast to our study, they used a benchtop WDXRF spectrometer instead of an EDXRF handheld system. As stated above, the WDXRF system is characterized by a higher accuracy. However, it can only measure one element at once resulting in lower measurement speed in comparison to an EDXRF system. Furthermore, in the study conducted by Roa-Espinosa et al. [19] samples were pressed to discs, which means an extra time-consuming and laboriously step for sample preparation. In comparison to loose powder, as used in our study, pressed discs show a higher and more uniform density, a more homogenous element distribution and a more uniform matrix structure, which all improves the precision and accuracy of XRF measurements [18]. Matsunami et al. [21] also used a benchtop WDXRF system with pressed discs and they observed slightly lower, but still excellent R^2^s for a mix of different manure types. Reasons might be that they analysed different manure types and the fact that they used 15 independent samples for their validation (i.e., not used for calibration).

The R^2^ values for poultry litter samples reported by Dao and Zhang [18] are a little bit higher than our R^2^ values (except for K) as they also used a benchtop instrument with pressed disc samples. The main reason, why Dao and Zhang [18] did not observe such a high degree of accordance as Roa-Espinosa et al. [19], is probably due to the fact that they used an EDXRF system. Koimiyama et al. [22] used the same benchtop EDXRF system and the disc preparation as sample pre-treatment. Although they examined three different manure types, their values are slightly better than those of Dao and Zhang [18]. However, the reason for that is not clear. Weindorf et al. [20] analysed composted dairy manure samples without any pre-treatment steps using a handheld XRF spectrometer, which is comparable to the one used in our study. Besides for Cu and Zn, Weindorf et al. [20] observed lower R^2^ values compared to our results, especially for K, which was very low with R^2^ = 0.14. Sapkota et al. [23] also used a handheld XRF spectrometer and achieved similar results. However, when they applied random forest regression to the raw XRF spectra, their R^2^ improved remarkably (especially for Mg from 0.14 to 0.90). Thus, there seems to be a great potential in chemometric models to improve the accuracy of element predictions in organic fertilizers using handheld XRF spectrometers.

## 4. Conclusions

Overall it can be concluded that XRF spectroscopy with handheld instruments in liquid manures and biogas digestates without special sample pre-treatment steps does not lead to reliable results (except for K), while the analysis of dried and ground samples is a promising procedure for on-farm analysis for the tested plant nutrients. Furthermore, recent studies revealed that specialized chemometric approaches like random forest might have a great potential to improve the reliability of handheld XRF spectrometers for measuring samples with a rather heterogeneous matrix like organic fertilizers. Thus, further research including the utilization of chemometric modelling and a higher number of samples is needed to confirm the results of this study.

However, working with XRF instruments is only allowed with a certified permission and recurring instructions due to the deleterious X-rays. Furthermore, a handheld XRF spectrometer cannot analyse nitrogen, which is the most important plant nutrient next to phosphor. Nevertheless, in addition to XRF analysis, a farmer or service provider could use simple on-farm quick tests for ammonium and total nitrogen.

## Figures and Tables

**Figure 1 sensors-21-03892-f001:**
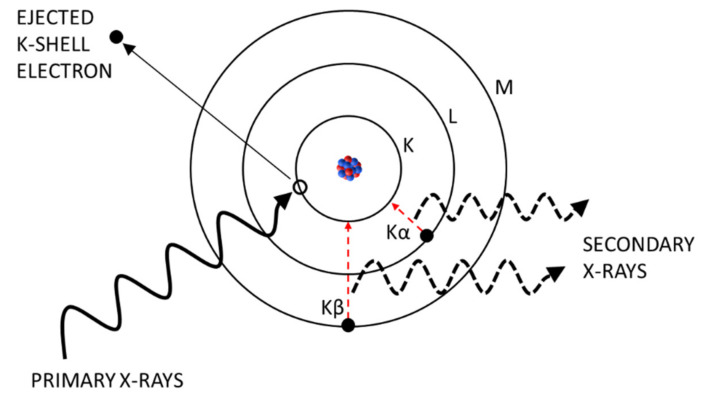
Physical mechanisms within an atom for X-ray-fluorescence.

**Figure 2 sensors-21-03892-f002:**
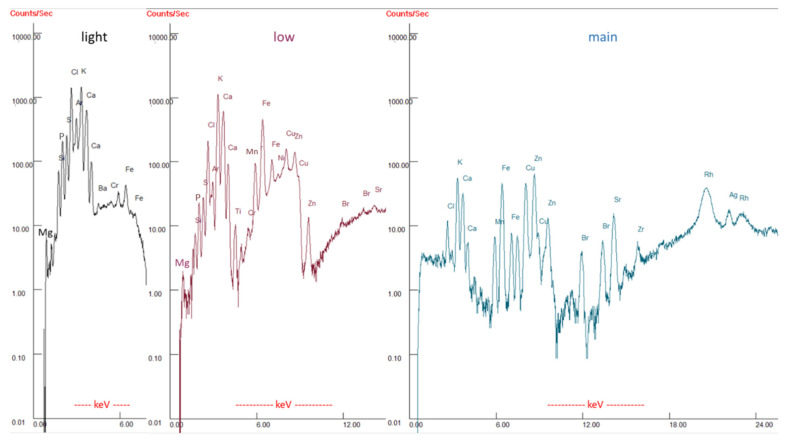
XRF fluorescence spectra of light, low, and main filter settings of a dried pig manure sample (log-scale for counts).

**Figure 3 sensors-21-03892-f003:**
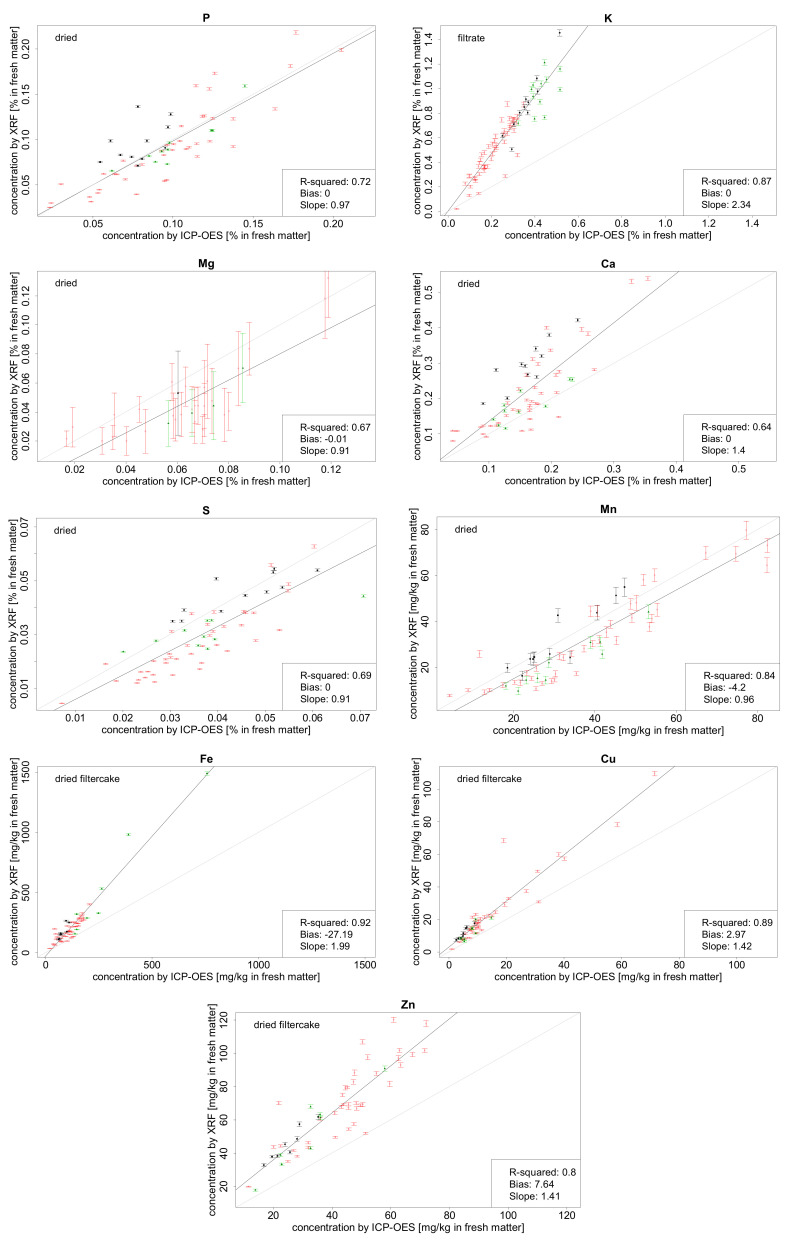
Best relationships between XRF and standard lab procedure based concentrations of P, K, Mg, Ca, S, Mn, Fe, Cu and Zn showing R^2^, bias, slope, XRF standard derivations, regression line (black), and 1:1 line (grey). For K, XRF concentrations are shown for filtrate samples, for Fe, Cu and Zn for dried filter residues and for P, Mg, Ca, S, and Mn for dried samples (red = pig, black = cattle, and green = digestate samples).

**Table 1 sensors-21-03892-t001:** Descriptive statistics for dry matter content (DM) and element concentrations in 62 liquid manure and biogas digestate samples, determined by the standard lab procedure (values are expressed on a fresh weight basis).

	DM	P	K	Mg	Ca	S	Mn	Fe	Cu	Zn
**Unit**	------------------------ % --------------------------						-------- mg/kg (ppm) -----			
**Min**	0.59	0.02	0.04	0.01	0.04	0.007	4.28	14.9	1.05	8.73
**Max**	11.0	0.21	0.52	0.12	0.35	0.071	82.4	758	71.5	76.6
**Average**	5.48	0.10	0.27	0.06	0.16	0.037	36.3	138	13.3	39.1
**Standard** **deviation**	2.64	0.04	0.11	0.02	0.06	0.012	17.2	116	14.5	16.4
**Coefficient of variation (%)**	48.2	37.2	41.8	35.4	36.4	31.28	47.2	84.1	108	41.0

Min: Minimum; Max: Maximum; DM: Dry matter.

**Table 2 sensors-21-03892-t002:** Coefficients of determination (R^2^) from linear regression between XRF and reference concentrations determined by the standard lab procedure (best values in bold letters) ^1^.

Sample Preparation	P	K	Mg	Ca	S	Mn	Fe	Cu	Zn
Original	0.28	0.83	-	0.17	0.02	-	0.52	0.58	0.40
Dried	**0.72**	0.77	**0.67**	**0.64**	**0.69**	**0.84**	0.82	0.86	0.71
Filtrate	0.33	**0.87**	-	0.08	0.04	-	0.68	0.16	0.37
Dried filter residues	0.56	0.65	0.33	0.55	0.66	0.81	**0.92**	**0.89**	**0.80**

^1^ The corresponding *p*-values confirm a strong significance for the regression models at a level of 10^−9^ to 10^−32^ for the best R^2^s of all target elements (data not shown).

**Table 3 sensors-21-03892-t003:** Pearson coefficients (r) for dry matter content (DM) and element concentrations in 62 liquid manure and biogas digestate samples, determined by the standard lab procedure. Correlation coefficients over 0.70 are highlighted in bold letters.

	DM	P	K	Mg	Ca	S	Mn	Fe	Cu	Zn
**DM**	1.00	0.46	**0.77**	0.57	0.44	0.69	0.39	0.32	−0.10	0.03
**P**	0.46	1.00	0.33	**0.82**	**0.76**	0.62	**0.85**	0.38	0.20	**0.73**
**K**	**0.77**	0.33	1.00	0.41	0.20	0.55	0.18	0.39	−0.23	−0.15
**Mg**	0.57	**0.82**	0.41	1.00	**0.73**	0.61	**0.78**	0.25	0.22	0.58
**Ca**	0.44	**0.76**	0.20	**0.73**	1.00	0.60	0.66	0.28	0.36	0.58
**S**	0.69	0.62	0.55	0.61	0.60	1.00	0.59	0.46	0.12	0.35
**Mn**	0.39	**0.85**	0.18	**0.78**	0.66	0.59	1.00	0.36	0.31	**0.75**
**Fe**	0.32	0.38	0.39	0.25	0.28	0.46	0.36	1.00	0.08	0.21
**Cu**	−0.10	0.20	−0.23	0.22	0.36	0.12	0.31	0.08	1.00	0.57
**Zn**	0.03	**0.73**	−0.15	0.58	0.58	0.35	**0.75**	0.21	0.57	1.00

**Table 4 sensors-21-03892-t004:** Comparison of R^2^ values found in the literature for selected elements (EDXRF = energy dispersive XRF detector; WDXRF = wavelength dispersive XRF detector); # 40 samples with 5 different dry matter contents, respectively.

Literature	Spectrometer-System	Number of Samples	Preparation and Sample Type	P	K	Mg	Ca	S	Mn	Fe	Cu	Zn
Dao and Zhang 2007 [18]	Benchtop EDXRF	71	pressed discs of poultry litter	-	0.849	0.840	0.900	0.727	0.902	-	0.959	0.900
Koimiyama et al. 2009 [22]	Benchtop WDXRF	31 (+15 for validation)	pressed discs of dairy-, pig-, and poultry manure compost	0.929	0.964	0.918	0.974	-	-	0.976	0.988	0.972
Matsunami et al. 2009 [21]	Benchtop EDXRF	122	pressed discs of dairy-, pig-, and poultry manure	0.976	0.955	0.861	0.980	-	0.970	0.986	0.994	0.988
Roa-Espinosa et al. 2016 [19]	Benchtop WDXRF	15	pressed discs of dairy manure	0.992	0.996	0.995	0.998	0.995	0.998	0.998	0.999	0.997
Sapkota et al. 2020 [23]	Handheld EDXRF	5 × 40 #	remoisturized dairy- and poultry manure	0.93	0.95	0.90	0.97	-	-	0.98	-	-
Weindorf et al. 2008 [20]	Handheld EDXRF	70	fresh dairy manure compost	-	0.140	-	0.510	-	0.669	0.667	0.946	0.811

## Data Availability

Available on request.

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
