# Peer review of "Determination of Nutrients in Liquid Manures and Biogas Digestates by Portable Energy-Dispersive X-ray Fluorescence Spectrometry"

_sensors, 2021, doi:10.3390/s21113892_

Round 1
Reviewer 1 Report
Review of the manuscript written by Michael Horf, Robin Gebbers, Sebastian Vogel, Markus Ostermann, Max-Frederik Piepel and Hans-Werner Olfs submitted to Sensors (Manuscript ID: sensors-1222972.
Title: Determination of Nutrients in Liquid Manures and Biogas Digestates by portable energy-dispersive X-ray Fluorescence Spectrometry.
The paper is devoted to the use of portable ED-XRF for determination of macro plant nutrients Mg, P, S, K, Ca and micro nutrients like Mn, Fe, Cu, Zn and Se in liquid manures and biogas digestates. The way of application is described and compared to the selected standard method, ICP-OES introduced according to DIN EN ISO 11885: 2009-09. Also the reference to the other norm (DIN EN 12880 : 2001-02) is given. However if references to these norms are recalled some methodological issues have to be explained. The introduction the first one includes the duties of users who would modify the area of its application:
“ISO 11885:2007 specifies a method for the determination of dissolved elements, elements bound to particles ("particulate") and total content of elements in different types of water (e.g. ground, surface, raw, potable and waste water) for the following elements: aluminium, antimony, arsenic, barium, beryllium, bismuth, boron, cadmium, calcium, chromium, cobalt, copper, gallium, indium, iron, lead, lithium, magnesium, manganese, molybdenum, nickel, phosphorus, potassium, selenium, silicon, silver, sodium, strontium, sulfur, tin, titanium, tungsten, vanadium, zinc and zirconium.
Taking into account the specific and additionally occurring interferences, these elements can also be determined in digests of water, sludges and sediments (for example, digests of water as specified in ISO 15587‑1 or ISO 15587‑2). The method is suitable for mass concentrations of particulate matter in waste water below 2 g/l. The scope of this method may be extended to other matrices or to higher amounts of particulate matter if it can be shown that additionally occurring interferences are considered and corrected for carefully. It is up to the user to demonstrate the fitness for purpose.”
To prove the usefulness of the selected ICP-OES method (the reference to the proposed ED-XRF spectroscopy) such demonstration of fitness for purpose is truly needed. I understand that Authors only referred to the so called standard method, but the proper use of ICP-OES has to be also indicated in this case. Even by the clear statement that it was checked and tuned according the expectations. The sentence written by Authors that ICP-OES is a method “known for its high precision without chemical interferences and a suitable conventional method to be selected for reference analysis because of its reliable and standardized procedures” (page 7 lines 277/280) is a huge simplification and overestimation of the analytical characteristics of the method. This sentence is not true even in eyes of ISO 11885:2007 which was recalled by Authors. Any statement about ICP-OES as the method with no interferences has to be cancelled from the manuscript and substituted by the estimation of possible interferences onto determination of the elements of interest. They (interferences) might be low but have to be mentioned in a discussion of the results.
Any information about the validation of the ED-XRF by analysis of Certified Reference Materials is needed. There are available Reference Materials which include all elements of interest and certificate with references to XRF measurements (for example NIST_695: https://www-s.nist.gov/srmors/certificates/695.pdf). Analysis of any CRM would allow for the real evaluation of the proposed methodology and estimation of its fitness to the described analytical aim.
The subject of described investigations is important for the state of the overall environment, therefore I would suggest to present the results according to the actual methodological and analytical standards. Any spectra might be shown while discussing about such problematic elements like Mg and the other light ones not easily detectable by portable XRF spectrometers.
The above remarks were the most crucial for the evaluation of the merit of the paper soundness.
I would suggest also:
- to use “time consuming” instead of “time-intensive”;
- to shorten introduction about the basics of XRF, the Figure 1 is one of the most popular among many descriptions of XRF and as so - it can be excluded;
- there are some inconsistency between the reported n=2 repetitions of the measurements and the total time: 3x30 s of the measurements, which suggests rather 3 repetitions. Please explain;
- there are too many tables with similar data, what is confusing and hard to be followed. The most illustrative ones are given in the form of Figure 2. Moreover some of the data from these tables are given in this Figure reported in the respective charts.
Reviewer 2 Report
This work shows interesting results for the reliability of using a handheld XRF for the determination of nutrients in different types of manure.
It could be acceptable for publication in Sensors but need some revisions. The citation of sources should be improved, you should use name of authors or summarize a method instead of just providing the reference number.
See underline text with comments in joined pdf file.

Author Response
The citation of sources should be improved, you should use name of authors or summarize a method instead of just providing the reference number.
-> done
missing bold markings
-> done
Thanks for reviewing!
Round 2
Reviewer 1 Report
Dear Editor,
I accept the manuscript and have no more comments to it.
Best regards,
Barbara Wagner